# Language Conditioned Spatial Relation Reasoning for 3D Object Grounding

**Shizhe Chen[1], Pierre-Louis Guhur[1], Makarand Tapaswi[2], Cordelia Schmid[1], Ivan Laptev[1]**

[1]Inria, École normale supérieure, CNRS, PSL Research University, [2]IIIT Hyderabad

`https://cshizhe.github.io/projects/vil3dref.html`

## Abstract

Localizing objects in 3D scenes based on natural language requires understanding and reasoning about spatial relations. In particular, it is often crucial to distinguish similar objects referred by the text, such as "the left most chair" and "a chair next to the window". In this work we propose a language-conditioned transformer model for grounding 3D objects and their spatial relations. To this end, we design a spatial self-attention layer that accounts for relative distances and orientations between objects in input 3D point clouds. Training such a layer with visual and language inputs enables to disambiguate spatial relations and to localize objects referred by the text. To facilitate the cross-modal learning of relations, we further propose a teacher-student approach where the teacher model is first trained using ground-truth object labels, and then helps to train a student model using point cloud inputs. We perform ablation studies showing advantages of our approach. We also demonstrate our model to significantly outperform the state of the art on the challenging Nr3D, Sr3D and ScanRefer 3D object grounding datasets.

## 1 Introduction

To carry out human instructions in the real world, robots should understand natural language and be able to ground mentioned objects in 3D environments. Following this objective, recent research is shifting from object grounding in 2D images [1, 2, 3, 4, 5, 6] to the 3D object grounding task [7, 8], where objects referred by a sentence should be localized in a 3D point cloud.

Language expressions often refer to objects by their relative spatial locations in 3D scenes. Figure 1 illustrates example scenes and corresponding sentences where object grounding requires disambiguation between objects of the same class. For instance, *"the backpack closest to the piano"* requires to compare relative distances among objects, while *"choose the door on the left when facing them"* requires to infer the correct viewpoint and understand relative directions. Such complexity and diversity of the spatial language makes 3D object grounding highly challenging.

Given the critical role of the spatial language, many existing methods attempt to model spatial relations for 3D object grounding. Early work [9, 10, 11, 12] explicitly build 3D visual graphs based on distances between objects and apply graph neural networks to learn relationships. However, since only nearest neighbors are considered in the graph, it remains difficult to infer relationships between distant objects such as *"farthest"*. More recently, transformer architectures [9, 13, 14, 15, 16] have been used, as they have the potential to learn relations between pairs of objects with a multi-head self-attention mechanism [17]. Enabling transformers to better understand 3D spatial relations expressed by natural language, however, remains an open research problem.

In addition, 3D object grounding suffers more from the scarcity of training data compared to its 2D counterpart. The success of 2D object grounding models [18, 19] can be largely attributed to large-scale datasets with image-text pairs [1, 2, 20, 21], whereas the limited amount of 3D scene-

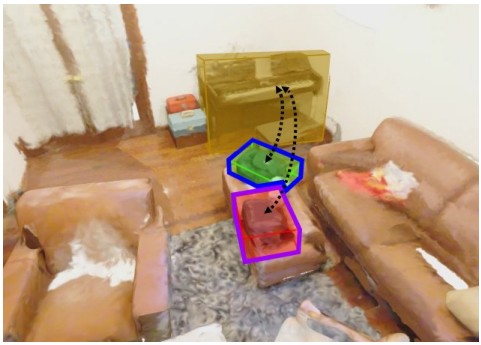 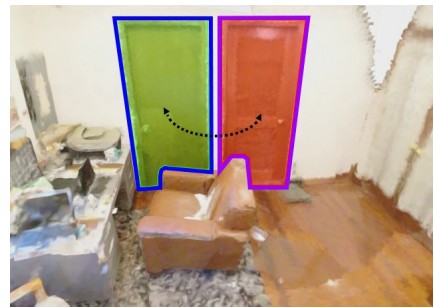

(a) The backpack closest to the piano.

(b) Of the two brown wooden doors, choose the door on the left when facing them.

Figure 1: Example sentences that refer to objects in 3D scenes. The green box denotes the ground-truth object, the blue box is the prediction from our model, and the purple one is from a baseline model without explicit spatial reasoning and knowledge distillation.

language pairs increases the difficulty of 3D object reasoning. To address this challenge, Yang *et al*. [14] propose to use 2D-3D alignments to assist the training of 3D models, but require additional high-quality 2D images and camera parameters which are not always available.

In this work, we propose a **Vi**sion-and-**L**anguage **3D Rel**ation reasoning model (ViL3DRel) to tackle the above issues in 3D object grounding. We design a new spatial self-attention layer for the transformer architecture to enhance 3D spatial understanding. This layer encodes relative distances and orientations for all pairs of objects, and explicitly learns spatial attention to capture spatial relations referred by the the language expressions. We propose a sigmoid softmax function to effectively fuse the spatial attention with standard self-attention. Rotation augmentation is used to further improve view invariant reasoning. To alleviate the negative effect of noisy object features, we further propose a teacher-student training approach. The teacher and student share the same model architecture, but use different inputs. The teacher uses ground-truth object labels, which enables to better learn the relation reasoning. The learned relation knowledge is distilled to the student taking point cloud features as input. Our code and models are available on the project webpage [22].

To summarize, our contributions are three-fold:

- We propose a ViL3DRel model for the 3D object grounding task. It uses a new spatial self-attention to explicitly encode pairwise 3D spatial relations in the transformer layer for better language conditioned spatial understanding.
- We employ a teacher-student training strategy, which facilitates the cross-modal learning of relations. The student with point cloud inputs benefits from the relation reasoning model of the teacher trained with ground-truth object label input.
- We evaluate our ViL3DRel approach on the challenging Nr3D, Sr3D [9] and ScanRefer [7] benchmarks. Our model significantly outperforms state-of-the-art approaches, with 9.3, 8.3 absolute gains on Nr3D and Sr3D given ground-truth object proposals, and 4.47 points on ScanRefer using the same detected object proposals compared to the previous work [16] .

## 2   Related Work

**2D and 3D Object Grounding.** The object grounding task aims to localize objects in 2D images or 3D point clouds given a sentence. Existing approaches can be categorized into two groups, namely one- and two-stage frameworks. The one-stage methods densely fuse the text features with patch- or point-level visual representations to directly regress the bounding box [18, 23]. These approaches are flexible to detect various objects given the input sentence. The two-stage methods adopt a *detection-then-matching* pipeline [1, 7, 8, 24], where the detection stage generates object proposals, and then the matching stage selects the best proposal according to the sentence. The decoupled object perception and cross-modal matching make the two-stage methods easier to analyze [15].

The benchmark datasets for 3D object grounding are ReferIt3D (Nr3D and Sr3D) [8] and Scan-Refer [7], which are all built upon scenes and object annotations in the ScanNet dataset [25]. Graph-based approaches [26, 27] are widely adopted in early works to infer spatial relations. The object graph is constructed by connecting each object with its top nearest neighbors [8, 10, 11] based

on Euclidean distance. Inspired by the success of transformers [17], recent works [9, 14, 15, 16, 23] have adopted transformers for 3D object grounding. BEAUTY-DETR [28] and 3D-SPS [23] are one-stage methods. However, most works follow the two-stage framework with pre-detected object proposals. LanguageRefer [15] converts the cross-modal task into a language modeling problem with predicted object labels. SAT [14] adopts a multimodal transformer and transfers 2D semantics to assist the training of the 3D model. Multi-view transformer [16] aggregates object representations from multiple views to improve view robustness. Perhaps most similar to our work, 3DVG-Transformer [13] proposes a coordinate-guided attention module to encode spatial distances among objects. In contrast, we introduce a spatial self-attention module, which explicitly encodes both relative distances and relative orientations among objects. It also conditions spatial relations on the language and presents a more effective strategy for attention fusion.

**Transformers in Vision-and-Language.** Transformer-based architectures have led to significant improvements in various vision-and-language tasks such as text-video retrieval [29], visual grounding [18, 30], image captioning [31], vision question-answering [32] and vision-and-language navigation [33, 34]. Most of the methods project textual and visual inputs into a sequence of tokens, and use multimodal transformers to learn cross-modal semantic relationships. While relative positional encoding (RPE) [35] has been explored mostly separately in vision and language transformers, we here integrate RPE with a cross-modal transformer and show its benefits for resolving spatial object relations referred by text.

**Knowledge Distillation.** Knowledge distillation [36] is typically used to compress a large network (teacher) into a compact model (student). The common objective is to let the student model mimic the soft logits of the teacher model. Beyer *et al*. [37] show that a consistent and patient teacher is essential in knowledge distillation. Jiao *et al*. [38] further demonstrate that the intermediate representations learned by the teacher are beneficial. In contrast to distilling knowledge from a heavy model to a light one, our teacher network first learns cross-modal object relations using ground-truth object labels. We then transfer this knowledge to the target student network that uses noisy inputs.

## 3 Method

Given a sentence $S$, the goal of 3D object grounding is to detect an object referred in $S$ by locating its 3D bounding box $B_T \in \mathbb{R}^6$ in a 3D point clouds $P_{scene}$. We assume $P_{scene} \in \mathbb{R}^{K \times 6}$ contains $K$ points each represented by XYZ coordinates and RGB values. We follow the *detection-then-matching* framework [7, 8, 13, 14, 16] for 3D object grounding and assume to be given a list of object proposals $(O_1, \cdots, O_N)$ obtained via automatic 3D instance segmentation or ground-truth annotations (depending on the evaluation setup). Each object $O_i$ is represented by a subset of $K_i$ points $P_i \subset P_{scene}, P_i \in \mathbb{R}^{K_i \times 6}$. Our work is focused on interpreting spatial relations and selecting the target object $O_T, T \in [1, N]$ among $N$ object proposals. We first present an overview of our ViL3DRel model in Section 3.1. We then introduce our language-conditioned spatial self-attention module and the teacher-student training approach in Sections 3.2 and 3.3 respectively.

### 3.1 Architecture Overview

Figure 2 shows an overview of our ViL3DRel model, consisting of four modules: text encoding, object encoding, multimodal fusion and a grounding head as described next.

**Text encoding.** Given the sentence $S$ with $M$ word tokens, we use a pre-trained BERT model [39] to encode $S$ into a sequence of word features $(s_{cls}, s_1, \cdots, s_M), s_i \in \mathbb{R}^d$, where $s_{cls}$ is a special classification token and $d$ is the dimensionality of the feature.

**Object encoding.** For each object $O_i$, we first normalize the coordinates of its point cloud $P_i$ into a unit ball, and then use PointNet++ [40] to compute the object feature $o_i^0 \in \mathbb{R}^d$. We also obtain the object center $c_i = [c_x, c_y, c_z] \in \mathbb{R}^3$ and the object size $z_i = [z_x, z_y, z_z] \in \mathbb{R}^3$ from object points $P_i$ as the mean and the spatial extent of $P_i$ respectively. We use a linear projection layer to obtain the absolute 3D location feature as $l_i = W_l[c_i; z_i] \in \mathbb{R}^d$.

**Multimodal fusion.** A stack of transformer layers [17] are applied to fuse the text and object modality features. Each transformer layer is composed of a spatial self-attention layer, a cross-attention layer and a feed-forward neural network (FFN). Our new spatial self-attention layer aims to improve the understanding of spatial relations among objects referred by the sentence. Assume $o_i^l$ is the input embedding for object proposal $O_i$ before the $i$-th layer, we first add it with its absolute 3D location feature $l_i$ to enhance the spatial information. The spatial self-attention then generates contextualized object representations $\hat{o}_i^l$. We will describe the proposed spatial self-attention in details in Section 3.2. The cross-attention layer takes $\hat{o}_i^l$ as queries and text features $(s_{cls}, s_1, \cdots, s_M)$ as keys and values

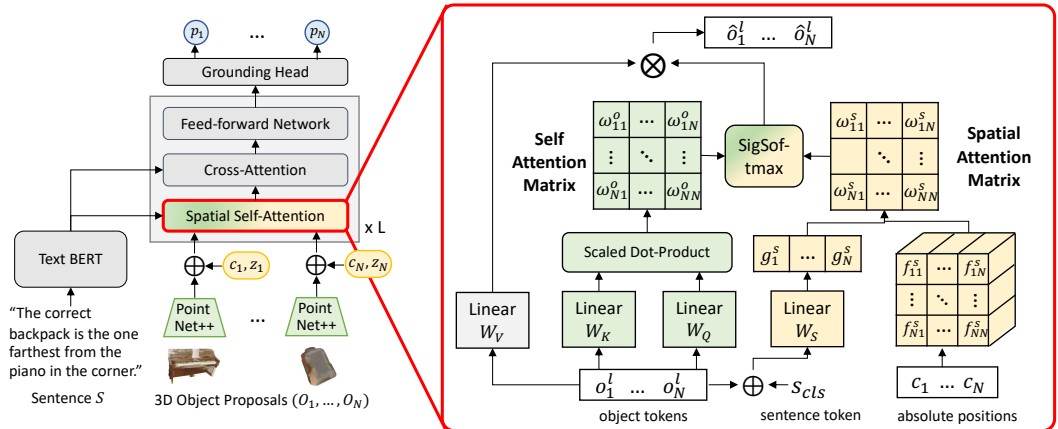

Figure 2: Left: overview of the ViL3DRel model; Right: language-conditioned spatial self-attention.

to learn cross-modal relations. The final FFN uses two fully connected layers to encode each output tokens after the attention layer. For detailed explanation of Transformer attention and FFN layers please refer to [17].

**Object grounding.** We use a two-layer feed-forward neural network as the object grounding head to predict the target object. Given the output embedding $o_i^L$ from the last multimodal fusion layer, the grounding head generates a scalar score for each object proposal $O_i$ and applies softmax function to obtain the probability $p_i$. The object proposal with the maximum probability is predicted as target.

## 3.2 Spatial Self-Attention

The standard self-attention relies on token embeddings to learn relations among tokens. However, the input embeddings of object proposals are mixed with semantic features and absolute 3D locations, making it difficult to accurately infer spatial relations among objects such as relative distances or orientations in Figure 1. As spatial relations among objects referred by language are important to distinguish object instances, we propose to inject a language conditioned spatial attention that explicitly captures pairwise spatial relations to complement the standard self-attention. We first describe the standard self-attention and then introduce our proposed spatial self-attention as illustrated in the right part of Figure 2.

Given the feature matrix $X \in \mathbb{R}^{N \times d}$ for $N$ object proposals, the self-attention mechanism first computes the query, key and value embeddings from $X$ as $Q = XW_Q, K = XW_K, V = XW_V$ respectively, where $W_Q, W_K, W_V \in \mathbb{R}^{d \times d_h}$ are learnable parameters and $d_h$ is the dimensionality of the output embedding. It then calculates attention weights given the query and key embeddings and aggregates the value embeddings as follows:

$$\Omega^o = \text{softmax}\left(\frac{QK^T}{\sqrt{d_h}}\right); \quad \text{SelfAttn}(Q, K, V) = \Omega^o V, \tag{1}$$

where $\Omega^o$ is an $N \times N$ attention matrix, whose elements $\omega_{ij}^o$ is the attention weight between the $i$-th and $j$-th object proposal. In order to capture more diverse relations, multi-head self-attention is used where each head computes an independent $\text{SelfAttn}(Q, K, V)$ and the outputs from all heads are concatenated.

To model spatial relations among objects, we propose to use explicit pairwise spatial features $f_{ij}^s \in \mathbb{R}^5, i, j \in [1, N]$. For each pair of objects $(O_i, O_j)$, we compute their Euclidean distance $d_{ij} = ||c_i - c_j||_2$ as well as horizontal and vertical angles $\theta_h, \theta_v$ of the line connecting object centers $c_i$ and $c_j$. The computation details of the angles are provided in the Section A of supplementary material. We then define the pairwise spatial feature $f_{ij}^s$ as:

$$f_{ij}^s = [d_{ij}, \sin(\theta_h), \cos(\theta_h), \sin(\theta_v), \cos(\theta_v)]. \tag{2}$$

We automatically generate a language conditioned weight $g_i^s$ to select relevant spatial relations for each object proposal $O_i$ as we mainly care about spatial relations described in the text for an object:

$$g_i^s = W_S^T(s_{cls} + o_i^l), \tag{3}$$

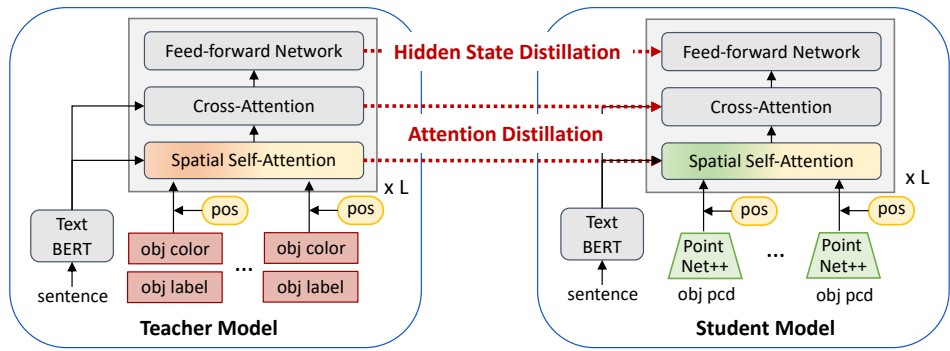

Figure 3: Teacher student learning with hidden state and attention distillation.

where $W_S \in \mathbb{R}^{d \times 5}$ is a learnable parameter and we omit the bias term for simplicity. We then define the spatial relevance for $(O_i, O_j)$ as:

$$\omega_{ij}^s = g_i^s \cdot f_{ij}^s. \tag{4}$$

We modulate the self-attention in Eq (1) with the above language conditioned spatial relevancy using the sigmoid softmax (sigsoftmax) fusion function:

$$\omega_{ij} = \frac{\sigma(\omega_{ij}^s) \exp(w_{ij}^o)}{\sum_{l=1}^{N} \sigma(\omega_{il}^s) \exp(w_{il}^o)}, \tag{5}$$

where $\sigma(\cdot)$ is the sigmoid function. In this way, we compute the new self-attention matrix $\Omega = [\omega_{ij}]_{N \times N}$ which explicitly considers 3D relative spatial locations, absolute spatial locations and object appearances in relation reasoning. We also use the multi-head mechanism to support different types of spatial relations and compute multiple spatial attentions to fuse with the standard self-attention. We concatenate the outputs from all heads in the spatial self-attention layer.

### 3.3 Teacher-Student Training

Incorrect estimation of object classes from $P_i$ can deteriorate the cross-modal alignment and spatial relation reasoning. To improve object grounding in point clouds previous work [14] relies on additional supervision in 2D images. Here, instead, we propose a new teacher-student training approach using no additional training data. The teacher and student models share the same transformer architectures except for the object encoding module. The teacher uses ground-truth object semantic features, while the student uses 3D point clouds. Since there is less cross-modal gap between the teacher's inputs and the sentence, the teacher model can excel at learning language relevant spatial relations among objects and aligning objects with the sentence. Such knowledge can then be distilled to the student model to improve its training. Figure 3 illustrates the teacher-student training pipeline. We describe the inputs of the teacher and the knowledge distillation objectives in the following.

**Teacher's inputs.** We use ground-truth class labels and dominant colors of the object as the object representations for the teacher model, since the color is the most widely used attributes in the sentence. Specifically, we encode the ground-truth object class label using pre-trained Glove word vectors [41]. To obtain the dominant colors of the object, we fit a Gaussian Mixture model on the RGB values of all points in the object, where the mixture component is set to 3. We linearly project the mean value in each component into a color embedding and use the mixture weights of the components to average all the color embeddings. The final object representation is the sum of the class label embedding and the averaged color embedding.

**Knowledge distillation objectives.** We transfer the attention weights and hidden states in the multimodal fusion module of the teacher model to the student model. The self-attentions learned by the teacher capture spatial relations among objects, and the cross-attentions measure cross-modal matching, which are both essential for 3D object grounding but hard to learn given noisy object features. We facilitate student learning by forcing it to mimic all the attention matrices $\Omega$ of the teacher using the following objective function:

$$\mathcal{L}_{attn} = \frac{1}{LH} \sum_{l=1}^{L} \sum_{h=1}^{H} \text{MSE}(\Omega_{lh}^S - \Omega_{lh}^T), \tag{6}$$

where $\Omega^S_.$ and $\Omega^T_.$ are attention weights in student and teacher model respectively, $\text{MSE}(\cdot)$ is the mean square error function, $L$ is the number of multimodal fusion layers and $H$ is the number of heads in self-attention. In addition to distilling the relation knowledge in attention weights, we also distill the output embeddings of the transformer layers:

$$\mathcal{L}_{hidden} = \frac{1}{LN} \sum_{l=0}^{L} \sum_{i=1}^{N} \text{MSE}(o_i^{lS} - o_i^{lT}). \tag{7}$$

**Training.** We follow the previous works [8, 7] to use multiple auxiliary losses in training the transformer model, including a 3D object grounding loss $\mathcal{L}_{og}$, sentence classification loss $\mathcal{L}_{sent}$ and two object classification losses $\mathcal{L}^u_{obj}$ and $\mathcal{L}^m_{obj}$. $\mathcal{L}_{sent}$ relies on $s_{cls}$ to predict the target object class from the sentence. $\mathcal{L}^u_{obj}$ and $\mathcal{L}^m_{obj}$ use unimodal object representation $o_i^0$ and multimodal fused object representation $o_i^L$ respectively to predict the classes for input object proposals. Details are in Section A of supplementary material. Therefore, the overall training objective is as follows:

$$\mathcal{L} = \mathcal{L}_{og} + \mathcal{L}_{sent} + \mathcal{L}^u_{obj} + \mathcal{L}^m_{obj} + \lambda_a \mathcal{L}_{attn} + \lambda_h \mathcal{L}_{hidden} \tag{8}$$

where $\lambda_a, \lambda_h$ are two hyper-parameters to balance the losses.

## 4 Experiments

### 4.1 Datasets

**Nr3D dataset [8]** contains 37,842 human-written sentences that refer to annotated objects in the 3D indoor scene dataset ScanNet [25]. The dataset includes 641 scenes with 511 (resp. 130) scenes for training (resp. validation). It covers 76 target object classes. The annotated sentences are designed to refer to objects with multiple same-class distractors in the scene. The sentences are split into "easy" and "hard" subsets in evaluation, where the target object in "easy" subset only contains one same-class distractor in the scene while it contains multiple ones in the "hard" subset. According to whether the sentence requires a specific viewpoint to ground the referred object, the dataset can also be partitioned into "view depedent" and "view independent" subsets.

**Sr3D dataset [8]** is constructed using templates to automatically generate sentences. The sentences only utilize spatial relations to distinguish objects of the same class. It has 1,018 training scenes and 255 validation scenes from ScanNet and 83,570 sentences in total. The dataset can be split in the same way as Nr3D during evaluation.

**ScanRefer dataset [7]** has 51,583 human-written sentences for 800 scenes in ScanNet. We follow the official split and use 36,665 and 9,508 samples for training and validation respectively. According to whether the target object is a unique object class in the scene, the dataset can be divided into a "unique" and a "multiple" subset.

### 4.2 Experimental Setting

**Evaluation Metrics.** We evaluate models under two evaluation settings. One uses ground-truth object proposals, which is the default setting in the Nr3D and Sr3D datasets. The metric is the accuracy of selecting the target bounding box among the proposals. The other setting does not provide ground-truth object proposals and requires the model to regress a 3D bounding box, which is the default setting for the ScanRefer dataset. The evaluation metrics are acc@0.25 and acc@0.5, which is the percentage of correctly predicted bounding boxes whose IoU is larger than 0.25 or 0.5 with the ground-truth.

**Implementation details.** For the model architecture, we set the dimension $d = 768$ and use 12 heads for all the transformer layers. The text encoding module is a three-layer transformer initialized from BERT [39], and the multimodal fusion module contains four layers. The object encoding module PointNet++ [40] samples 1024 points for all the objects. These architecture parameters are the same as in previous work [14, 16] to ensure a fair comparison. We first train the PointNet++ for object classification on ScanNet, which achieves 61.9% accuracy on the validation set. Its weights remain fixed during the following training steps. Rotation augmentation is used to increase the viewpoint invariance. The hyper-parameters in the loss function Eq (8) are set to $\lambda_a = 1$ and $\lambda_h = 0.02$. We train the model with a batch size of 128 and a learning rate of 0.0005 with warm-up of 5000 iterations and cosine decay scheduling. The AdamW algorithm [42] is used in the optimization. We train for 50 epochs for the teacher model and 100 epochs for the student model on Nr3D and ScanRefer datasets. The training epochs are reduced to half on the Sr3D dataset, as it is larger and easier to converge. All models are trained on a single NVIDIA RTX A6000 GPU.

Table 1: Grounding accuracy (%) on the Nr3D dataset with ground-truth object labels. Dist stands for Distance; Ort for Orientation; MHA for multi-head spatial attention; RotAug for Rotation Augmentation; sigs for sigsoftmax in Eq (5); and '-' means not applicable.

| | Spatial Relation Reasoning | | | | Rot Aug | Color | Overall | ViewDep | ViewIndep |
| | Dist | Ort | MHA | Fusion | | | | | |
|---|---|---|---|---|---|---|---|---|---|
| R1 | - | - | - | - | × | × | 53.5 | 51.4 | 54.6 |
| R2 | - | - | - | - | × | ✓ | 55.1 | 53.8 | 55.8 |
| R3 | - | - | - | - | ✓ | ✓ | 62.4 | 58.3 | 64.5 |
| R4 | ✓ | × | ✓ | sigs | ✓ | ✓ | 66.0 | 53.8 | 72.0 |
| R5 | × | ✓ | ✓ | sigs | ✓ | ✓ | 71.3 | 68.5 | 72.6 |
| R6 | ✓ | ✓ | × | sigs | ✓ | ✓ | 67.7 | 65.2 | 69.0 |
| R7 | ✓ | ✓ | ✓ | bias | ✓ | ✓ | 55.4 | 46.8 | 59.6 |
| R8 | ✓ | ✓ | ✓ | ctx | ✓ | ✓ | 56.4 | 50.8 | 59.1 |
| R9 | ✓ | ✓ | ✓ | sigs | ✓ | ✓ | **74.4** | **71.3** | **75.9** |

## 4.3 Ablation Studies

We carry out extensive experiments on the Nr3D dataset to demonstrate the effectiveness of our proposed spatial self-attention and teacher-student training. The ablations on the ScanRefer dataset are provided in Section B of supplementary material.

### 4.3.1 Spatial Relation Reasoning

We first evaluate the proposed spatial self-attention using ground-truth object labels, which decouples the object perception from spatial relation reasoning. Table 1 presents results on the Nr3D dataset.
**Baselines.** Our baseline R1 is the teacher model in the left part of Figure 3 that excludes object color, rotation augmentation and the proposed spatial self-attention. This baseline achieves similar performance to the state-of-the-art LanguageRefer [15] (overall accuracy: 54.3%) which also uses ground-truth object labels. Row R2 shows that adding color information helps object grounding as sentences often contain color attributes. The rotation augmentation brings additional gains and improves accuracy from 55.1% (R2) to 62.4% (R3). It is more effective for view independent sentences (+8.7%) than for view dependent sentences (+4.5%) because rotation augmentation mainly improves view invariance. We consider R3 as a strong baseline and use it to demonstrate improvements of our proposed model.
**Pairwise spatial features: distance vs. orientation.** Rows R4-R5 in Table 1 compare different pairwise spatial features defined in (2). R4 only uses pairwise distances, while R5 only uses pairwise orientations. We can see that pairwise distances are beneficial for view-independent sentences which contain distance-related spatial relations such as *"next to"* and *"farthest from"*, but do not improve the performance for view-dependent sentences. In contrast, pairwise orientations significantly boost the performance for view-dependent sentences (+10.2%). This shows that explicitly encoding the relative orientations facilitates the learning of view dependent spatial relations such as *"in front of"* and *"to the left of"*. The relative distance and orientation features are also complementary. Their combination achieves the best performance as shown in our full model (R9).

Table 2: Comparison of pairwise spatial feature computation methods.

| | Overall |
|---|---|
| object center | 74.4 |
| bottom center | 74.4 |
| boxes + MLP | 57.4 |

Table 3 provides a more systematic analysis of R3-5 and R9 in Table 1 to show the contribution of spatial relation modeling to different types of sentences. We categorize sentences into four groups according to whether the sentence describes spatial relations in terms of distances or orientations. We can see that the explicit pairwise distance modeling contributes most to the distance-only sentences but has little influence on samples with orientation-related sentences. On the other hand, the pairwise orientation modeling can significantly improve orientation-related sentences by 10.5%. Combining both pairwise distance and orientation modeling achieves the best performance on all categories.

**Pairwise spatial feature computation.** We compute pairwise distances and angles using the coordinates of object centers and design the spatial features as shown in Eq (2). In Table 2, we further

Table 3: Performance breakdown of models in Table 1 by spatial relation types: *Dist(Ort) only* which only contains distance(orientation) descriptions; *Dist & Ort* which contains both distance and orientation descriptions; and the *Others* which do not contain spatial relation descriptions.

| Dist | Ort | Overall | *Dist only* | *Ort only* | *Dist & Ort* | *Others* |
|------|-----|---------|-------------|------------|--------------|----------|
| × | × | 62.4 | 63.5 | 61.2 | 57.7 | 63.9 |
| ✓ | × | 66.0 | 72.6 | 58.9 | 55.1 | 68.8 |
| × | ✓ | 71.3 | 73.8 | 71.7 | 67.7 | 69.1 |
| ✓ | ✓ | **74.4** | **77.8** | **74.0** | **69.1** | **72.6** |

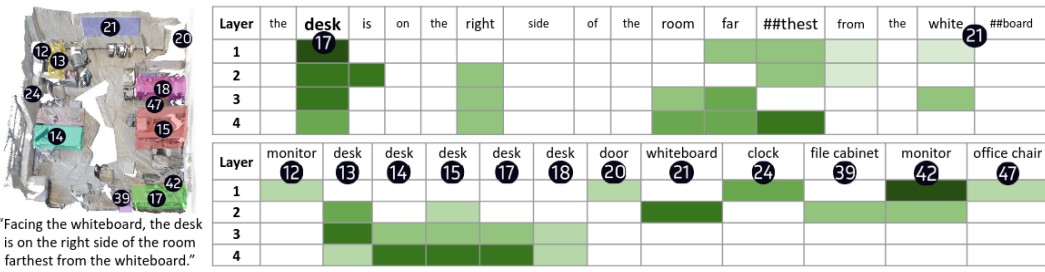

Figure 4: An example of the learned cross-attention (upper) and self-attention (bottom) at different transformer layers in the teacher model. The query is the target object proposal (desk 17). The darker color denotes higher attention weights from the query to the keys.

compare the proposed method with two variants to show its effectiveness. The first variant uses the bottom center of objects to compute pairwise vertical angles which ignores the height of objects and could be more accurate to measure vertical relations. The second variant concatenates bounding boxes of two objects and uses a multi-layer perceptron (MLP) to learn pairwise spatial relations. We can see that using the bottom center achieves similar performance as using the object center, which suggests that the object center is sufficient to capture the vertical spatial relations of objects in the textual descriptions. The learnable MLP, however, achieves much worse performance than our approach. This indicates that it is challenging to implicitly learn pairwise spatial relations and the proposed spatial features designed by domain knowledge are beneficial.

**Multi-head spatial attention.** R6 in Table 1 uses single-head attention for the spatial relevance $\omega_{ij}^s$ in (5) but keeps multi-head for the standard attention weight $\omega_{ij}^o$. Compared to the full model R9 with the multi-head spatial attention, such single-head spatial attention achieves significantly worse performance. This suggests that the multi-head is beneficial for learning spatial relations.

**Attention fusion.** We use sigmoid softmax function in (5) to aggregate the spatial attention and the standard self-attention weights. We compare the proposed fusion mechanism with standard relative positional encoding methods [35] in vision transformers: bias mode and contextual mode. The bias mode adds the spatial attention weight as a bias term in the standard self-attention, while the contextual mode further considers the interaction with queries by injecting the pairwise spatial features into the key embeddings. From the results in R7 and R8, we can see that the standard relative positional encoding methods fail to model 3D spatial relations among objects; they even perform worse than the strong baseline (R3), which doesn't rely on relative positional encoding. The proposed sigmoid softmax function is by far more effective to modulate the standard self-attention with spatial relations.

Table 4: Performance comparison of different textual encoders.

| Encoder | #Layers | Overall |
|---------|---------|---------|
| BERT | 3 | **74.4** |
| | 6 | 73.7 |
| | 9 | 73.2 |
| | 12 | 52.3 |
| Glove+GRU | 3 | 45.7 |

**Text encoding.** We follow recent works [14, 16] to use the first three layers of BERT textual encoder, while [11] uses weaker textual encoder (Glove+GRU). To compare these different textual encoders, Table 4 provides results of our proposed models under the setting in Table 1. We can see that more BERT layers do not lead to better performance. The reason might be that the representations from higher BERT layers are less generalizable to different domains. Moreover, more layers are more prone

Table 6: Grounding accuracy (%) on Nr3D and Sr3D datasets with ground-truth object proposals.

| Method | Nr3D | | | | | Sr3D | | | | |
|---|---|---|---|---|---|---|---|---|---|---|
| | Overall | Easy | Hard | View Dep | View Indep | Overall | Easy | Hard | View Dep | View Indep |
| ReferIt3D [8] | 35.6 | 43.6 | 27.9 | 32.5 | 37.1 | 40.8 | 44.7 | 31.5 | 39.2 | 40.8 |
| ScanRefer [7] | 34.2 | 41.0 | 23.5 | 29.9 | 35.4 | - | - | - | - | - |
| TGNN [10] | 37.3 | 44.2 | 30.6 | 35.8 | 38.0 | - | - | - | - | - |
| InstanceRefer [11] | 38.8 | 46.0 | 31.8 | 34.5 | 41.9 | 48.0 | 51.1 | 40.5 | 45.4 | 48.1 |
| FFL-3DOG [12] | 41.7 | 48.2 | 35.0 | 37.1 | 44.7 | - | - | - | - | - |
| 3DVG-Trans [13] | 40.8 | 48.5 | 34.8 | 34.8 | 43.7 | 51.4 | 54.2 | 44.9 | 44.6 | 51.7 |
| TransRefer3D [9] | 42.1 | 48.5 | 36.0 | 36.5 | 44.9 | 57.4 | 60.5 | 50.2 | 49.9 | 57.7 |
| LanguageRefer [15] | 43.9 | 51.0 | 36.6 | 41.7 | 45.0 | 56.0 | 58.9 | 49.3 | 49.2 | 56.3 |
| SAT [14] | 49.2 | 56.3 | 42.4 | 46.9 | 50.4 | 57.9 | 61.2 | 50.0 | 49.2 | 58.3 |
| 3D-SPS [23] | 51.5 | 58.1 | 45.1 | 48.0 | 53.2 | 62.6 | 56.2 | 65.4 | 49.2 | 63.2 |
| Multi-view [16] | 55.1 | 61.3 | 49.1 | 54.3 | 55.4 | 64.5 | 66.9 | 58.8 | 58.4 | 64.7 |
| ViL3DRel (Ours) | **64.4** | **70.2** | **57.4** | **62.0** | **64.5** | **72.8** | **74.9** | **67.9** | **63.8** | **73.2** |

to overfit and harder to optimize. The pretrained BERT model achieves much better performance compared to the GRU model trained from scratch.

### 4.3.2 Teacher-Student Training

The teacher model with ground-truth object labels learns well how to perform spatial relation reasoning given the input sentence. Figure 4 provides a qualitative example of the learned cross- and self-attention weights at all multimodal fusion layers in the teacher model. For the cross-attention weights, we can see that the teacher first aligns the object proposal with the correct words in the text, and then gradually shifts its attentions to relation words to perform spatial relation reasoning. For the self-attention weights, in the first layer, the teacher mostly attends to nearby objects to be aware of the context; then in the second

Table 5: Grounding accuracy (%) on the Nr3D dataset with ground-truth object proposals.

| | init. | $\mathcal{L}_{attn}$ | $\mathcal{L}_{hidden}$ | Overall |
|---|---|---|---|---|
| Teacher | | - | | 74.4 |
| Student | × | × | × | 58.1 |
| | ✓ | × | × | 62.6 |
| | × | ✓ | × | 63.6 |
| | × | × | ✓ | 62.1 |
| | × | ✓ | ✓ | **64.4** |

layer, it focuses more on the reference object mentioned in the sentence (*e.g.*, the whiteboard); finally, it concentrates on the same-class distractors to distinguish them. Such reasoning steps are promising to ease the training of the student model. In the following, we evaluate the proposed teacher-student training to distill relational knowledge from the teacher to a student with point cloud inputs extracted from the ground-truth object proposals.

Table 5 presents the performance of different student models. The first row in the student block does not use any knowledge from the teacher model. In the second row, we use the weights in the teacher model to initialize the student model since the teacher and student share the same architecture except for the input object representations. Such simple weight initialization already facilitates the training of the student, and achieves 4.5% gains. The attention distillation in (6) and hidden state distillation in (7) are both beneficial to improve the student model compared to the first row without knowledge distillation. It is more effective to distill the relation knowledge in the attention weight matrices. The combination of $\mathcal{L}_{attn}$ and $\mathcal{L}_{hidden}$ is helpful and achieves the best performance. We empirically find that combining weight initialization and the two distillation losses performs similar to using the distillation losses alone.

### 4.4 Comparison with State-of-the-Art Methods

Table 6 compares our ViL3DRel model with state-of-the-art methods on Nr3D and Sr3D datasets. All the compared works use ground-truth object proposals, but no ground-truth labels. Our model achieves significant improvements over the previous best method [16], with 9.3% and 8.3% absolute gains on Nr3D and Sr3D respectively. We outperform [13] by even larger margins on the two datasets. This work also use spatial information, but doesn't use a spatial attention module.

Table 8: Grounding accuracy (%) on ScanRefer with detected object proposals. VN and PG denote the VoteNet and PointGroup models pretrained on 18 classes of the ScanNet dataset, while Optimized denotes end-to-end training of the object detector on the ScanRefer dataset.

| Method | Det | Unique | | Multiple | | Overall | |
|---|---|---|---|---|---|---|---|
| | | acc@0.25 | acc@0.5 | acc@0.25 | acc@0.5 | acc@0.25 | acc@0.5 |
| ScanRefer [7] | | 65.00 | 43.31 | 30.63 | 19.75 | 37.30 | 24.32 |
| TGNN [10] | Opti- | 68.61 | 56.80 | 29.84 | 23.18 | 37.37 | 29.70 |
| 3DVG-Trans [13] | mized | 77.16 | 58.47 | 38.38 | 28.70 | 45.90 | 34.47 |
| 3D-SPS [23] | | 81.63 | 64.77 | 39.48 | 29.61 | 47.65 | 36.42 |
| Non-SAT [14] | VN [45] | 68.48 | 47.38 | 31.81 | 21.34 | 38.92 | 26.40 |
| SAT [14] | VN [45] | 73.21 | 50.83 | 37.64 | 25.16 | 44.54 | 30.14 |
| InstanceRefer [11] | PG [43] | 77.45 | 66.83 | 31.27 | 24.77 | 40.23 | 32.93 |
| Multi-view [16] | PG [43] | 77.67 | 66.45 | 31.92 | 25.26 | 40.80 | 33.26 |
| ViL3DRel (Ours) | PG [43] | **81.58** | **68.62** | **40.30** | **30.71** | **47.94** | **37.73** |
| UpperBound | PG [43] | 88.63 | 74.47 | 78.82 | 60.37 | 80.64 | 62.98 |

Table 7 and Table 8 present results on ScanRefer dataset with ground-truth object proposals and detected object proposals, respectively. For fair comparison, we use a pre-trained PointGroup [43] model to generate object proposals, which is trained on 18 classes in ScanNet. To be noted, the upper block in Table 8 optimizes the detection stage or utilizes single-stage pipeline. These methods use more data to train object proposals, so the comparison to these methods is not entirely fair. There is a large gap between the ground-truth and detected proposals (over 12% for our method), because the detected proposals might not include the target or reference objects in the sentence. However, our ViL3DRel model still outperforms the state of the art in both settings. Though our proposed spatial relation module can also be plugged into transformer-based 3D object detectors [44] to improve the 3D object proposal generation, we leave it to future work. Figure 1 compares our ViL3DRel model and a baseline without spatial self-attention and teacher-student training. We present more qualitative comparisons in Section C of supplementary material.

Table 7: Grounding accuracy (%) on ScanRefer with ground-truth object proposals.

| | Overall |
|---|---|
| ReferIt3D [8] | 46.9 |
| Non-SAT [14] | 48.2 |
| SAT [14] | 53.8 |
| ViL3DRel (Ours) | **59.8** |

## 5 Conclusion

In this work, we propose a ViL3DRel model for 3D object grounding. It contains a newly designed spatial self-attention module to improve language conditioned spatial relation reasoning in the transformer layer, which explicitly considers relative distances and orientations among objects. A teacher-student training approach is further proposed to transfer relation knowledge from a teacher with ground-truth object labels to a student with point cloud inputs. The proposed model significantly outperforms the state of the art on Nr3D, Sr3D and ScanRefer datasets.

Beyond the application of 3D object grounding, the general approach of incorporating priors into a self-attention layer for a transformer to combine multimodal input can be of wide interest. Since our model belongs to the two-stage framework, it is limited by imperfect object proposals in the first detection stage. We also did not explore to explicitly extract object orientations to more accurately estimate pairwise spatial relations as the automatic object poses prediction remains a challenging problem. In addition, the evaluation can suffer from the fact that existing datasets do not represent a rich diversity of environments though we carried out extensive ablations to mitigate the problem. This work has minimal ethical, privacy and safety concerns.

## Acknowledgments and Disclosure of Funding

This work was granted access to the HPC resources of IDRIS under the allocation 101002 made by GENCI. It was funded in part by the French government under management of Agence Nationale de la Recherche as part of the "Investissements d'avenir" program, reference ANR19-P3IA-0001 (PRAIRIE 3IA Institute), the ANR project VideoPredict (ANR-21-FAI1-0002-01) and by Louis Vuitton ENS Chair on Artificial Intelligence.

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
