# Supplementary Material for Language Conditioned Spatial Relation Reasoning for 3D Object Grounding

**Shizhe Chen[1], Pierre-Louis Guhur[1], Makarand Tapaswi[2], Cordelia Schmid[1], Ivan Laptev[1]**

[1]Inria, École normale supérieure, CNRS, PSL Research University, [2]IIIT Hyderabad

https://cshizhe.github.io/projects/vil3dref.html

## A  Implementation Details

### A.1  Computing horizontal and vertical angles between two objects

For two objects A and B, assume that the coordinates of the object center are $(x_a, y_a, z_a)$ and $(x_b, y_b, z_b)$. We can compute the Euclidean distance between A and B as $d$. The horizontal angle $\theta_h$ is $\arctan2(\frac{y_b - y_a}{x_b - x_a})$ and the vertical angle $\theta_v$ is $\arcsin(\frac{z_b - z_a}{d})$. In other terminology, the horizontal angle corresponds to the azimuth direction while looking from point A to B, while the vertical angle corresponds to the elevation.

### A.2  Rotation augmentation

When performing rotation augmentation, we rotate the whole point cloud by different angles. Specifically, we randomly select one angle among $[0°, 90°, 180°, 270°]$.

### A.3  Training losses

We use auxiliary losses $L_{sent}$ and $L^*_{obj}$ following previous works [1, 2, 3, 4, 5] which have shown to be beneficial for the performance. They are all cross entropy losses. $L_{sent}$ is to predict the target object class from the sentence. $L^*_{obj}$ is to predict the object class for each object token. When training the teacher model, all the losses are used. When training the student model, the $L^u_{obj}$ does not influence model weights since the pointnet is fixed.

## B  Additional Results

### B.1  Ablations Studies on ScanRefer Dataset

In the main paper, we provide ablation studies on the Nr3D dataset. Here, we further evaluate our method on the ScanRefer dataset to demonstrate the effectiveness of the proposed spatial self-attention and teacher-student training.

Table 1 presents the grounding accuracy using the ground-truth object labels on the ScanRefer dataset. By comparing R1-R3, we observe that the color information and rotation augmentation is beneficial to provide a stronger baseline. The proposed spatial self-attention (R9) significantly improves the strong baseline (R3), with the absolute gain of 4.79%. However, different from the Nr3D dataset, the sentences in the ScanRefer dataset are less focused on spatial relations and contain more attribute descriptions such as colors and shapes to refer to target objects. Therefore, the overall performance gain from R3 to R9 is smaller compared to the Nr3D dataset. The rows R4 and R5 analyze the contributions of relative spatial features to the performance. We can see that both relative distances and orientations outperform the baseline in R3, and that the relative distances are more important

Table 1: Grounding accuracy (%) on the ScanRefer dataset with ground-truth object labels. Dist stands for Distance; Ort for Orientation; MHA for Multi-Head spatial Attention; RotAug for Rotation Augmentation; sigs for sigsoftmax in Eq (5); and '-' means not applicable.

| | Spatial Relation Reasoning | | | | Rot Aug | Color | Overall |
| | Dist | Ort | MHA | Fusion | | | |
|---|---|---|---|---|---|---|---|
| R1 | - | - | - | - | × | × | 56.35 |
| R2 | - | - | - | - | × | ✓ | 57.80 |
| R3 | - | - | - | - | ✓ | ✓ | 59.10 |
| R4 | ✓ | × | ✓ | sigs | ✓ | ✓ | 63.87 |
| R5 | × | ✓ | ✓ | sigs | ✓ | ✓ | 60.28 |
| R6 | ✓ | ✓ | × | sigs | ✓ | ✓ | 63.39 |
| R7 | ✓ | ✓ | ✓ | bias | ✓ | ✓ | 59.16 |
| R8 | ✓ | ✓ | ✓ | ctx | ✓ | ✓ | 58.79 |
| R9 | ✓ | ✓ | ✓ | sigs | ✓ | ✓ | **63.89** |

in the ScanRefer dataset. Their combination achieves the best performance in R9. The multi-head attention is also beneficial comparing R6 and R9. In R7 and R8, we compare the proposed sigmoid softmax function with the other two common relative position encoding methods. Our proposed fusion method is most effective to exploit both the spatial attention and the standard self-attention weights.

Table 2: Grounding accuracy (%) on the ScanRefer dataset with ground-truth object proposals.

| | init. | $\mathcal{L}_{attn}$ | $\mathcal{L}_{hidden}$ | Overall |
|---|---|---|---|---|
| Teacher | | - | | 63.89 |
| Student | × | × | × | 57.50 |
| | ✓ | × | × | 57.81 |
| | × | ✓ | × | 59.67 |
| | × | × | ✓ | **59.89** |
| | × | ✓ | ✓ | 59.80 |

Table 2 compares models using ground-truth object proposals on the ScanRefer dataset. The first row in the student model block does not use any knowledge from the teacher model which uses ground-truth object labels as inputs. Initializing from the weights in the teacher only brings a slight improvement as shown in the second row. However, our proposed knowledge distillation achieves over 2% boost compared to the baseline in the first row. Both attention weights and hidden states are beneficial to train a better student model with noisy object features. The hidden state distillation slightly outperforms attention distillation which is different from the results on the Nr3D dataset. The reason could be that the spatial relations are mentioned more frequently in the Nr3D dataset while object attributes are more typical in the ScanRefer dataset.

## B.2   Robustness to Random Seeds

We use the same random seed of 0 for all the experiments. In the following, we verify the robustness of the proposed ViL3DRel model by measuring the average and standard derivations of the grounding accuracy under 5 different random seeds. From the results in Table 3, we can see that the proposed model is stable with low deviations across different random seeds.

Table 3: The robustness of the proposed ViL3DRel model with respect to different random seeds (%).

| | Nr3D | Sr3D | ScanRefer |
|---|---|---|---|
| seed=0 | 64.4 | 72.8 | 37.7 |
| 5 seeds | $63.8 \pm 0.5$ | $72.8 \pm 0.2$ | $37.6 \pm 0.2$ |

## C   Qualitative Results

Figure 1 compares predictions by our proposed ViL3DRel model and the baseline model without the spatial self-attention and knowledge distillation. We can see that our model is better at object perception (left example) while reasoning about different types of spatial relations such as relative distances (middle example) and orientations (right example).

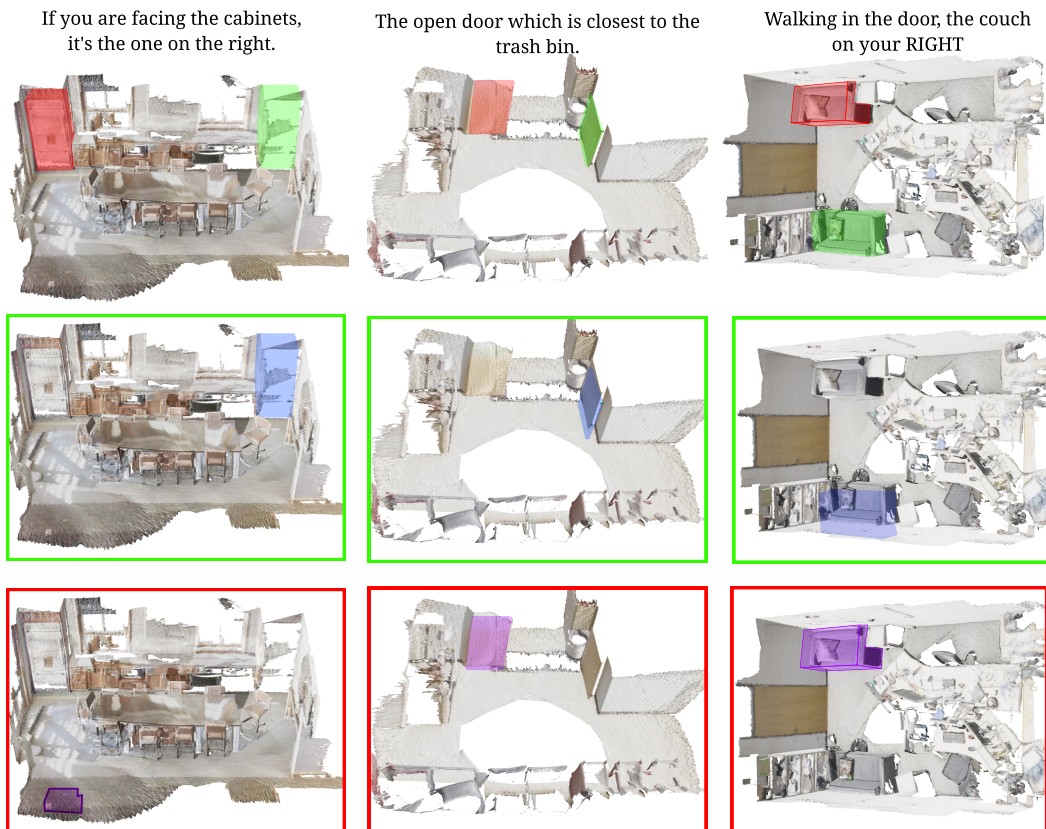

Figure 1: Examples on the Nr3D dataset. The first row presents the ground-truth where the green boxes denote the target object and the red boxes denote distractor objects of the same class. The second row shows the predictions (blue boxes) from our proposed ViL3DRel model. The third row shows the predictions (purple boxes) from a baseline model without spatial self-attention and knowledge distillation.

In Figure 2, we further provide some failure cases of the ViL3DRel model. Example on the left is missing object proposals for the outlet and television. Since our model belongs to the two-stage method, its performance is highly dependent on the quality of object proposals in the first stage. The middle example suggests that the current model still suffers from recognizing fine-grained attributes from the point clouds. The right example further requires reasoning about object orientations, which might be hard to estimate from noisy point clouds.

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

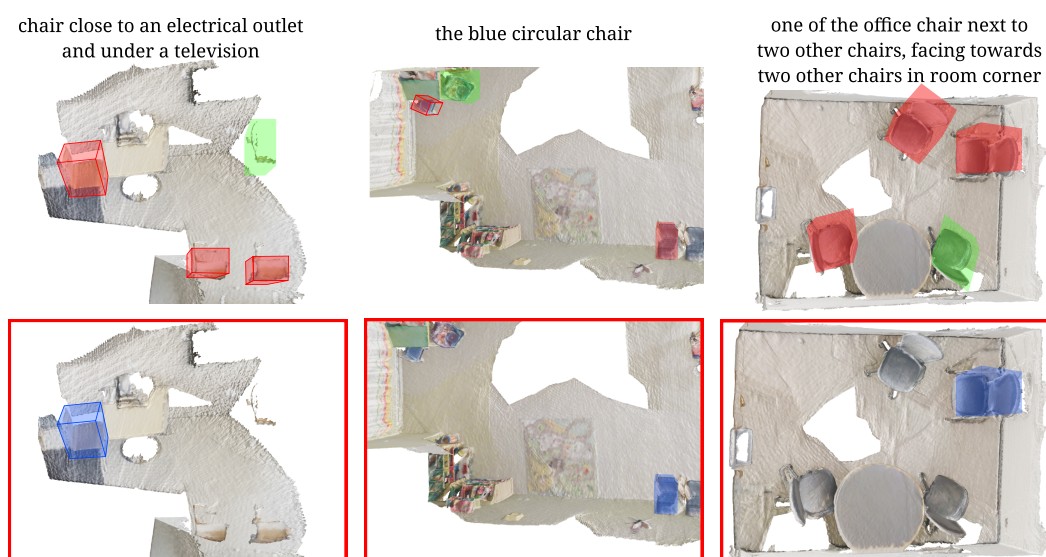

Figure 2: Failure cases of the proposed ViL3DRel model. The first row presents the ground-truth where the green boxes denote the target object and the red boxes denote distractor objects of the same class. The second row shows the predictions (blue boxes) from our proposed ViL3DRel model.