# OpenReview forum: "Language Conditioned Spatial Relation Reasoning for 3D Object Grounding"
_NeurIPS.cc/2022/Conference — NeurIPS 2022 Accept_

### Official Review · Reviewer_cFMa · 2022-07-11

**Rating:** 6
**Confidence:** 2
**Soundness:** 3 good
**Presentation:** 3 good
**Contribution:** 3 good

**Summary:**

This paper proposes a language-conditioned transformer model for localizing objects in 3D scenes and reasoning about relative spatial distances based on natural language description. The model is based on the transformer architecture and includes spatial attention layers and multi-head design to learn and reason about relative distances and orientations among multiple objects. The spatial attention output is fused with standard self-attention by the sigmoid softmax function proposed by the authors. The model is trained using a teacher-student approach accompanied by rotation augmentation to facilitate learning under scarce data and to produce reasonable inference with point cloud inputs. The final model is tested on object grounding tasks Nr3D, Sr3D, and ScanRefer3D, and outperformed state-of-the-art models.

**Questions:**

As mentioned in the weaknesses above, clarifying the language conditioned weight, BERT text feature input, and auxiliary tasks would be helpful. In addition, it would be helpful to include more details about the cross-attention computation in the model.

**Limitations:**

The object proposal in the two-stage pipeline is critical because it constrains the hypothesis space for cross-modal matching. Thus the prospect of improvement in the object proposal stage would be beneficial.

**Strengths And Weaknesses:**

Strengths:
1. This paper addresses a challenging and interesting task of 3D object reasoning based on natural language description, with a focus on distinguishing similar objects referred by the text using different spatial relationships.
2. Using multi-head attention, the proposed language-conditioned transformer model effectively learns to capture and reason about different spatial relationships among objects in the self-attention layers.
3. The model is systematically evaluated on multiple datasets and achieves strong performance, and the learned attention is clearly demonstrated in Figure 4.
4. The ablation study is carefully designed and shows the value of different components the authors introduce in the paper.

Weaknesses:
1. In equation 3, it is not very intuitive what g (language conditioned weight) means. What is the role of the classification token s_cls being an input in the conditioning?
2. Related to the previous point, in Figure 2 (left), the Text BERT serves as input to both the spatial self-attention and cross-attention. It is not clear if the self-attention takes the text features (s_1, …, s_M) as input or only takes the s_cls token.
3. In line 189, the authors state that the teacher model’s input object representation is the sum of object class label embedding and the averaged color embedding. Does the sum introduce any difficulty during training, especially when it requires the model to use object appearance (such as color) rather than spatial relationships to locate objects?
4. It is nice to see the visualization of the teacher model in Figure 4. It will be compelling if the authors show that the student model attention layers behave similarly.
5. The loss function in equation 8 is very complicated. There is little detail about the auxiliary losses and the rationale for including them in the objective.
6. The final model only marginally outperforms 3D-SPS, according to Table 5. Although the authors state that the comparison is not fair due to detection optimization or single-stage pipeline, it would be helpful to reflect on how to improve the current detected object proposal.

---

> ### Author Response · Authors · 2022-08-02
> **Response to Reviewer cFMa**
>
> We thank the reviewer for acknowledging our contributions and for constructive comments to our work.
>
> **Q1: In equation 3, it is not very intuitive what $g$ (language conditioned weight) means. What is the role of the classification token $s_{cls}$ being an input in the conditioning?**
>
> $g$ in Eq (3) is a weight to be multiplied with pairwise spatial feature $f$ in Eq (4). We care more about the spatial relations mentioned in the language for an object. For example, in the sentence “the bag closest to the piano”, we would like to obtain a weight $g$ for the object “bag” that is able to filter the distance relation. Therefore, we use the global sentence token $s_{cls}$ and the object token $o_i$ to predict the weight.
>
> **Q2: Related to the previous point, in Figure 2 (left), the Text BERT serves as input to both the spatial self-attention and cross-attention. It is not clear if the self-attention takes the text features $(s_1, …, s_M)$ as input or only takes the $s_{cls}$ token.**
>
> The spatial self-attention layer only applies self-attention over object tokens. The text features $(s_1, …, s_M)$ are not the input. Only the $s_{cls}$ token is used as in Eq (3).
>
> **Q3: In line 189, the authors state that the teacher model’s input object representation is the sum of object class label embedding and the averaged color embedding. Does the sum introduce any difficulty during training, especially when it requires the model to use object appearance (such as color) rather than spatial relationships to locate objects?**
>
> The sum operation is standard in existing works [17]. Given the high capacity of the model, it is possible to learn a manifold where the sum operation is effective to fuse the two embeddings.
>
> **Q4: It is nice to see the visualization of the teacher model in Figure 4. It will be compelling if the authors show that the student model attention layers behave similarly.**
>
> We observe that the attention layers in the student model gradually converge to the teacher’s behavior thanks to the knowledge distillation losses in Eq (8). We will provide the training curves of the distillation losses in the updated version which measure the similarities of attention weights between teacher and student.
>
> **Q5: The loss function in equation 8 is very complicated. There is little detail about the auxiliary losses and the rationale for including them in the objective.**
>
> The auxiliary losses $L_{sent}$ and $L^*_{obj}$ are standard in previous works [7,9,13,14,16] which have shown to be beneficial for the performance. They are all cross entropy losses. $L_{sent}$ is to predict the target object class from the sentence. $L^*_{obj}$ is to predict the object class for each object token. We will clarify it in the updated version.
>
> **Q6: The final model only marginally outperforms 3D-SPS, according to Table 5. Although the authors state that the comparison is not fair due to detection optimization or single-stage pipeline, it would be helpful to reflect on how to improve the current detected object proposal.**
>
> For fair comparison with previous two-stage methods, we use the PG object detector [42]. The state-of-the-art works on 3D object detection/segmentation are also based on transformer architectures. Our proposed spatial relation module can be applied in such transformer architectures to enhance the contextual modeling. Moreover, the textual information can be applied to address the few-shot or zero-shot object detection. However, we leave this to future work.
>
> **Q7: It would be helpful to include more details about the cross-attention computation in the model.**
>
> The cross-attention layer follows the standard transformer work [17]. The queries are the object tokens, while keys and values are the textual tokens. We will provide more details of the cross-attention computation in the supplementary material. We will also release the codes and trained models if the paper is accepted.

---

### Official Review · Reviewer_KM3K · 2022-07-11

**Rating:** 6
**Confidence:** 3
**Soundness:** 4 excellent
**Presentation:** 4 excellent
**Contribution:** 3 good

**Summary:**

This paper proposes a new method for 3D object grounding with a transformer that combines input of a BERT text encoder and a Point++ object encoder to match the detected object proposals with the input sentence description.

The key insight here is that explicitly encoding spatial relations between detected objects significantly help with spatial reasoning. Based on this observation, the method incorporates a spatial self-attention layer that computes spatial relevance from language inputs and the explicitly defined pairwise object relations (of relative distance and angle). This spatial relevance is then used to weight the self attention between objects.

Combining this novel spatial reasoning layer with teacher-student knowledge that distills knowledge from a teacher network with access to ground truth labels and color of objects, the proposed method is able to achieve significantly better performance than prior works on multiple benchmarks, where reasonable control of variables are attempted. Thorough ablation are also performed to justify the importance of each of the novel design choices.


**Questions:**

### Design Choices
- What is the motivation of computing Euclidean distance from the centroids, instead of between the bounding boxes? It seem to me that distance between bounding boxes might describe proximity between objects more unambiguously as opposed to centroid-to-centroid distances.
- Similarly, it seems that the vertical angles between objects might be very sensitive to height of the objects. Would it be better to decouple the impact of dimension e.g. by computing this angle based on the center of the bottom face?
- I wonder what's the benefit of hardcoding spatial relations as opposed to, say, using a MLP to automatically compute spatial relations from the bounding boxes of the objects, probably in their local coordinate frame?
- I imagine many text descriptions are probably related with the room architecture itself. Would there be benefit in also trying to encode object-room relations explicitly?
- I know its mentioned in the supplementary that determining the orientations from noisy inputs is challenging. Still, it appears to be that most natural language descriptions of directions (left, right, etc) is highly depending on the orientation of the objects, so the model will need to reason about orientation in some way. Have you tried to more explicitly handle orientations when defining and encoding the spatial relations?  If, does inclusion of them degrades the performance because of the ambiguity? If not, what are some possible ways to potential handle such cases?

### Evaluation
- As mentioned, some discussion of alternative ways to design the spatial relation layer might be helpful, but definitely not necessary.


**Limitations:**

Only one limitation that is common to all method of this class is mentioned towards the end. More discussion of limitation specific to this method would help.
Qualitative failure cases are provided in the supplementary, with analysis. One example (Supp Figure 2, right) shows a weakness specific to this method regarding object orientation.

I agree that there is no apparent negative social impact.

**Strengths And Weaknesses:**

First and foremost: I have not worked on the problem studied in this paper and was not familiar with the related works until reviewing this paper. So this is more of a common-sense review judging from the technical soundness and quality of evaluation, as well as quick skim through of some of the relevant works discussed. I'll defer the judgment about the magnitude of contribution to reviews who are more familiar with this problem.

That being said, I like this work in general: it has a very clear motivation (improving spatial reasoning for 3D object grounding) and proposes an effective solution that clearly outperforms relevant prior works. My concern about this paper is largely around the magnitude of the contribution and if that have significant impact on future works. Again, I don't think I am really qualified to judge about this, but based on other points below, I lean towards accepting this paper.

### Strengths:
+ Simple yet important key observation about the importance of explicitly encoding pairwise spatial relations, solid design of a novel module based on this observation.
+ A generalizable teacher-student training method that can be extended to other methods for the problem studied.
+ Exceptional results: significantly outperforms relevant prior works on multiple benchmarks and even outperforms works with more assumptions on input.
+ Extensive ablations validating each of the design choices.
+ Good writing quality, easy to follow.

### Weaknesses:
- Magnitude of contribution might be slightly limited since it follows a rather standard pipeline. The novel part of this paper can, to some extent, be seen training tricks / inductive biases to help the model better picking up certain information (spatial relation) that is important to the task.
- Following the previous point: it might be argued that the techniques introduced in this network might be limiting eventually if solutions emerge to better capture the spatial relations from the object attributes alone.
- Ablations only justify that each of the component help with the final result, without discussion of alternative method that might achieve the same goal. I think this understandable though - should not expect a work that explores an idea to exhaustively study the design space.

---

> ### Author Response · Authors · 2022-08-02
> **Response to Reviewer KM3K**
>
> We thank the reviewer for the constructive comments to our work.
>
> **Q1: The magnitude of contributions might be slightly limited since it follows a standard pipeline. The novel part of this paper can, to some extent, be seen as training tricks / inductive biases to help the model better picking up certain information (spatial relation) that is important to the task.**
>
> We contend that spatial relation modeling is important for 3D object grounding. As pointed out by Reviewer yLyV, “The described spatial self-attention mechanism is to my knowledge a novel contribution in this area”.
> We would like to emphasize that the proposed teacher-student training method is also important. The main contribution of the previous SAT [13] is to use additional 2D images to assist the training in 3D object grounding. Our teacher-student training does not rely on such additional supervisions and outperforms existing approaches.
>
> **Q2: It might be argued that the techniques introduced in this network might be limiting eventually if solutions emerge to better capture the spatial relations from the object attributes alone.**
>
> We contend that improving object attributes is complementary with spatial relation modeling. In Table 1, although we use the ground-truth object labels and colors which are strong object representations, the proposed spatial relation modeling still achieves significant improvements over the baseline.
>
> **Q3: Ablations only justify that each of the components help with the final result, without discussion of alternative methods that might achieve the same goal. I think this is understandable though - should not expect a work that explores an idea to exhaustively study the design space.**
>
> Though we do not extensively explore the design space, we carefully select and compare against two representative approaches to model spatial information in Table 1 (bias and ctx [34]) and show that our fusion strategy outperforms them.
>
> **Q4: What is the motivation of computing Euclidean distance from the centroids, instead of between the bounding boxes? It seems to me that distance between bounding boxes might describe proximity between objects more unambiguously as opposed to centroid-to-centroid distances.**
>
> We compute the distance from the centroids as it is simple and effective as shown in Table 1. The distances of centroids are sufficient to  distance related phrases (see keywords in the reply to Reviewer yLyV).  It would be more complicated and require more ad-hoc design to compute distances between bounding boxes.
>
> **Q5: Would it be better to computer vertical angle based on the center of the bottom face?**
>
> We follow the suggestion and use the bottom center of objects to compute the pairwise vertical angles. The result is in the second row of the table below. Using the bottom center achieves the same performance as using the object center, which suggests that our original design is sufficient to capture the vertical spatial relations of objects.
>
> |  | Overall |
> |---|:---:|
> | object center | 74.4 |
> | bottom center | 74.4 |
> | object bounding boxes + MLP | 57.4 |
>
> **Q6: I wonder what's the benefit of hardcoding spatial relations as opposed to, say, using a MLP to automatically compute spatial relations from the bounding boxes of the objects, probably in their local coordinate frame?**
>
> The result of computing the spatial relations from bounding boxes with a MLP is presented in the last row of the table above. We can see that it achieves much worse performance than our designed spatial relation features. This indicates that it is challenging to implicitly learn pairwise spatial relations and our designed features are beneficial.
>
> **Q7: I imagine many text descriptions are probably related with the room architecture itself. Would there be benefit in also trying to encode object-room relations explicitly?**
>
> We observed that the object-room relations are relatively simple such as “in the corner of the room” etc. Such information is encoded in the absolute positions of the objects.
>
> **Q8: Have you tried to more explicitly handle orientations when defining and encoding the spatial relations?**
>
> We have not explored object orientations in our work as the corresponding ground truth was not available for datasets considered in our work and as the automatic estimation of object poses remains a challenging problem. We believe using object orientations could bring further improvements and we leave this direction for future work.
>
> **Q9: More discussion of limitations specific to this method would help.**
>
> We could discuss more specific limitations of our model in the updated version such as missing explicit object orientations.

---

### Official Review · Reviewer_9xAT · 2022-07-11

**Rating:** 6
**Confidence:** 4
**Soundness:** 4 excellent
**Presentation:** 4 excellent
**Contribution:** 4 excellent

**Summary:**

This paper focuses on improving spatial relationship reasoning in the task of 3D object referring expression comprehension. The key ideas are a dedicated spatial attention mechanism integrated into the basic transformer architecture, and a teacher-student training mechanism. The paper significantly outperforms existing baselines on ReferIt and ScanRefer datasets.

**Questions:**

L159: Can you be more specific about the horizontal and vertical angles?

L165: Why do you use sigsoftmax, in contrast to, for example, softmax？

L235: Is there any reference on using only the first 3 layers of BERT to encode the sentence? Why don't use all layers? Also, it seems that some baselines (InstanceRefer) use (perhaps) weaker textual encoders (GloVe+GRU), can you provide an ablation study on the textual encoder module?

L239/L205: If the pointnet is pretrained and fixed, why are you adding $L_{obj}^u$?

L240: The author should explain the details of the rotation augmentation.


**Limitations:**

What are the limitations of this paper?

**Strengths And Weaknesses:**

The paper is clearly motivated and well written. The model is based on a straightforward transformer architecture, and integrates spatial attention models. Although I would argue that the color feature and spatial relationship feature used in the paper is a little ad-hoc but it is simple and effective. I have only very few questions and suggestions about the weakness of the paper.

Perhaps the biggest weakness of the paper is that, based on my understanding, the base architecture (e.g., the R1 model in Table 1) is not an existing baseline model. Thus, the authors are both changing the basic architecture (c.f., InstanceRefer and Multi-View) and add-ons (teacher-student, spatial attention). To better understand the effectiveness of the proposed modules, the authors should consider integrating the presented techniques with other baselines too.

---

> ### Author Response · Authors · 2022-08-02
> **Response to Reviewer 9xAT**
>
> We thank the reviewer for detailed and constructive comments. We address the raised points in the following.
>
> **Q1: The base architecture (R1 in Table 1) is not an existing baseline model. To better understand the effectiveness of the proposed modules, the authors should consider integrating the presented techniques with other baselines too.**
>
> In Table 1, we carry out extensive ablations of the newly proposed methods on top of our baseline model (R1). The results demonstrate the effectiveness of our proposed methods in a fair comparison setting.
> Moreover, our base architecture (R1) is similar to existing works that are based on transformers such as LanguageRefer [15], 3DVG-Transformer [13], SAT [14] and Multi-view transformer [16]. We argue that the slight differences in the base architectures do not have a large influence on the performance. For example, LanguageRefer [15] reported the performance under the same setting as our Table 1 which uses ground-truth object labels on the Nr3D dataset. We can see that the performance of the two models is also similar. Therefore, we believe that the improvement of our model compared to previous transformer-based methods is unrelated to the base architecture.
>
> |  | Overall | ViewDep | ViewIndep |
> |---|:---:|:---:|:---:|
> | R1 Tab 1 (Our baseline) | 53.5 | 51.4 | 54.6 |
> | LanguageRefer [15] | 54.3 | 49.1 | 56.8 |
>
> **Q2: L159: Can you be more specific about the horizontal and vertical angles?**
>
> For two objects A and B, assume that the coordinates of the object center are $(x_a, y_a, z_a)$ and $(x_b, y_b, z_b)$. We can compute the Euclidean distance between A and B as $d$. The horizontal angle $\theta_h$ is $arctan2(\frac{y_b - y_a}{x_b - x_a})$ and the vertical angle $\theta_v$ is $arcsin(\frac{z_b - z_a}{d})$. In other terminology, the horizontal angle corresponds to the azimuth direction while looking from point A to B, while the vertical angle corresponds to the elevation.
>
> **Q3: L165: Why do you use sigsoftmax, in contrast to, for example, softmax？**
>
> We tried different ways to encode the spatial information. The R7 and R8 in Table 1 indeed use the softmax function. Empirically, the softmax variants perform worse than our proposed sigsoftmax fusion. The reason might be that sigsoftmax can more aggressively modify the original self-attention weights with spatial information.
>
> **Q4: L235: Is there any reference on using only the first 3 layers of BERT to encode the sentence? Why don't you use all layers? Also, it seems that some baselines (InstanceRefer) use (perhaps) weaker textual encoders (GloVe+GRU), can you provide an ablation study on the textual encoder module?**
>
> We use the first three layers of BERT following the setup in more recent works such as SAT [13] and Multi-view transformer [16]. Here we provide additional results of different textual encoders under the setting in Table 1. We can see that more BERT layers do not lead to better performance. The pretrained BERT model achieves much better performance than GRU model trained from scratch. We will include these ablations in the final version.
>
> |  | Overall |
> |---|:---:|
> | BERT (first 3 layers) | 74.4 |
> | BERT (first 6 layers) | 73.7 |
> | BERT (first 9 layers) | 73.2 |
> | BERT (all 12 layers) | 52.3 |
> | Glove+GRU (3 layers) | 45.7 |
>
> **Q5: L239/L205: If the pointnet is pretrained and fixed, why are you adding $L^u_{obj}$?**
>
> These auxiliary losses are the same for the teacher and student models, so we simply add the term $L^u_{obj}$ in the final loss. It will not influence the pointnet in the student model. We will clarify it in the updated version.
>
> **Q6: L240: The author should explain the details of the rotation augmentation.**
>
> The rotation augmentation is to rotate the whole point clouds by different angles. Specifically, we randomly select one from four angles [0°, 90°, 180°, 270°].
>
> **Q7: What are the limitations of this paper?**
>
> We briefly mentioned the limitation in the conclusion section. Our model is a two-stage framework and thus is limited by imperfect object proposals in the first detection stage. As kindly mentioned by Reviewer yLyV, there also exists an inherent dataset bias.

---

> > ### Comment · Reviewer_9xAT · 2022-08-07
> > **Thank You For The Response**
> >
> > Thank you for the detailed response. The discussions about baselines and the language encoders are helpful. Since there is a huge performance gap between different language encoders, I would suggest the authors add this discussion to the main paper. Overall, I am satisfied with the author's responses and would keep my rating as acceptance.

---

### Official Review · Reviewer_yLyV · 2022-07-12

**Rating:** 7
**Confidence:** 3
**Soundness:** 4 excellent
**Presentation:** 4 excellent
**Contribution:** 3 good

**Summary:**

The paper proposes a method for language grounding in 3D scenes (i.e. detecting a specific 3D object instance based on an input natural text describing the object in the context of the scene).  The method is based on a transformer architecture, with the key novelty lying in a spatial self-attention mechanism and a teacher-student distillation training setup.

The method is evaluated on the ScanRefer and ReferIt3D datasets, with comparisons against ablations to demonstrate the value of method components, and also against baselines taken from prior work to demonstrate the relative improvement in 3D object grounding.  The experiments report grounding accuracy and show measurable improvements over prior work.

**Questions:**

I would like the authors to address the identified weakness above, at least in terms of the broad strokes of how the approach performs for input text exhibiting different degrees of spatial relation disambiguation.

**Limitations:**

The conclusion mentions the two-stage nature of the approach as a limitation (i.e. that object detections are assumed to be given, and may be imperfect).  The conclusion also briefly states that the presented work only has minimal potential negative societal impact.  I would agree with this statement given the domain (3D interiors), but I would still point out that inherent dataset biases may misrepresent the rich diversity of environments in which people around the world may live.

**Strengths And Weaknesses:**

Strengths
+ The paper is clearly motivated in the need to better model 3D spatial relations in the context of the language grounding problem. The described spatial self-attention mechanism is to my knowledge a novel contribution in this area
+ The exposition is overall quite clear, and the experiments present a thorough spectrum of ablations that do a good job of analyzing the impact of different method components (in particular, the attention to contrasts between view-dependent vs view-independent text inputs is appreciated)

Weaknesses
- Given the big focus on spatial relations, I would have liked to see a more systematic analysis of the results dependent on the need for attending to spatial relations to disambiguate the specific object instance that the input text describes (as illustrated well in Fig 1).  For example, a comparison of overall performance on sentences like those seen in Fig 1 vs sentences that do not use spatial relations to disambiguate would offer more detailed evidence that the proposed architecture is indeed better able to ground objects in such cases.  In addition, a breakdown by spatial relation types (involving distance, involving orientation, and involving both) would be quite informative.

---

> ### Author Response · Authors · 2022-08-02
> **Response to Reviewer yLyV**
>
> We thank the reviewer for providing constructive comments. We are happy that the reviewer acknowledged our novel contribution in 3D spatial relation modeling and thorough experiments.
>
> **Q1: More systematic analysis of results. 1) A comparison of overall performance on sentences with and without spatial relations. 2) A breakdown by spatial relation types (involving distance, involving orientation, and involving both).**
>
> In order to carry out more systematic analysis, we categorize sentences into four groups: 1) *Dist only* which only contains distance descriptions; 2) *Ort only* which only contains orientation descriptions; 3) *Dist & Ort* which contains both distance and orientation descriptions; and 4) the *others* which do not contain spatial relation descriptions.
>
> As the existing datasets do not have such categorization, we first manually select keywords among top words in the dataset that are relevant to distances or orientations, and consider a sentence has distance or orientation descriptions if it has any of those keywords. Specifically, the distance related keywords are: 'far', 'farther', 'farthest', 'close', 'closer', 'closest', 'next', 'near', 'nearest', 'beside', 'between', 'middle', and the orientation related keywords are: 'front', 'behind', 'back', 'right', 'left', 'leftmost', 'rightmost', 'above', 'under'.
> The data percentages are presented in the first row of the table below.
>
> We provide additional analysis for the results of different models in Table 1 in the following table. Comparing R3 and R4 we can see that the explicit pairwise distance modeling contributes most to the distance only sentences but has little influence on orientation related sentences. The pairwise orientation modeling instead can dramatically improve orientation related sentences by 10.5%. Combining both pairwise distance and orientation modeling in R9 achieves the best performance on all categories. Thanks to the reviewer for suggesting this experiment, we will include these results in our updated version.
>
> |  | Dist | Ort | Overall | Dist only | Ort only | Dist & Ort | Others |
> |:---:|:---:|:---:|:---:|:---:|:---:|:---:|:---:|
> | data percentage |  |  | 100 | 33.8 | 27.4 | 9.8 | 29.0 |
> | R3 | x | x | 62.4 | 63.5 | 61.2 | 57.7 | 63.9 |
> | R4 | ✔️ | x | 66.0 | 72.6 | 58.9 | 55.1 | 68.8 |
> | R5 | x | ✔️ | 71.3 | 73.8 | 71.7 | 67.7 | 69.1 |
> | R9 | ✔️ | ✔️ | 74.4 | 77.8 | 74.0 | 69.1 | 72.6 |
>
> **Q2: Limitation on inherent dataset biases.**
>
> Thank you for pointing out this limitation. We agree that the existing datasets might not represent a rich diversity of environments as they are all built upon the ScanNet dataset with about 1500 scenes. We will mention the possible biases in the updated version.

---

### Meta-Review · Area_Chair_7Mp6 · 2022-08-30

**Recommendation:** Accept
**Confidence:** Certain

**Metareview:**

Reviewers where in agreement that the method and manuscript are strong and provide a valuable connection between 3D perception and language.

The evaluation suffers somewhat from the fact that there are no good datasets targeted at this problem. Authors mitigate this by performing a thorough evaluation with numerous ablations/alternatives that demonstrate their method works as intended.

Beyond just this application, the general approach taken of incorporating priors about a problem domain into a self-attention layer for a Transformer to combine multimodal input is timely and will be of wide interest.

I encourage the authors to update the manuscript and include the additional experiments and results they produced for reviewers.


**Award:**

No

---

### Decision · Program_Chairs · 2022-09-14

Accept